# Unclogging the Ears: Nonstop Languaging as Autotheory in Art and Academia

**Antrianna Moutoula**

Independent Researcher, Spreeuwenpark 11H, 1021 GS Amsterdam, The Netherlands;
antrianna.moutoula@gmail.com

**Abstract:** This text emerges from the autotheoretical performance practice of nonstop languaging, developed during my Master Studies at HOME OF PERFORMANCE PRACTICES, ArtEZ University of the Arts. My ongoing artistic research is enacted through this practice and proposes a writing of the self that is not focused on recalling facts or narrating stories, but rather on tracing my thoughts in real time through language (languaging) and witnessing them simultaneously with another person. I perform autotheory by merging methods of articulating autobiography (carrying the self in language) with methods of forming and digesting theory. For this Special Issue, I created a new work in which nonstop languaging enters the framework of an academic paper. The autotheoretical work was developed through a series of radio performances at radio WORM, followed by a period of artistic research on transcription and citational practices. The result is an overload of words, thoughts, citations, experiences, theories, and memories that seek their own linearity. The practice of nonstop languaging contributes to the current artistic and academic discourse on autotheoretical modes of working with language, particularly within contemporary art, and further afield. This article invites readers to engage with an expanded view of autotheory in practice, and suggests that, by encouraging the shaping of an audience of engaged readers/listeners, autotheory can offer a space in which the confinements of knowledge production and dissemination within artistic academic discourse can be renegotiated.

**Keywords:** nonstop languaging; autotheory; performance; necessary other; engaged readers; citational practices; contemporary art; knowledge production; feminism





## 1. Introduction

> *"If woman has always functioned 'within' the discourse of man, a signifier that has always referred back to the opposite signifier which annihilates its specific energy and diminishes or stifles its very different sounds, it is time for her to dislocate this 'within,' to explode it, turn it around, and seize it; to make it hers, containing it, taking it in her own mouth, biting that tongue with her very own teeth to invent for herself a language to get inside of."* (Cixous 1976, p. 887)

Hannah Arendt, in *The Life of the Mind*, wonders "whether thinking and other invisible and soundless mental activities are meant to appear or whether in fact they can never find an adequate home in the world" (Arendt 1978, p. 23). Gertrude Stein sees streams of consciousness as a linguistic tool which can transform reality (Sitrin 2013–2014, p. 111). Language poet Lyn Hejinian states that one is always "thinking about reality" as "reality is all there is" (Hejinian 2000, p. 8). Influenced by those ideas, in my artistic research, I position the appearance of thinking and, specifically, the stream of thought as a crucial element in the production of autotheory. I explore streams of consciousness through the performance practice which I have named nonstop languaging (see Figure 1). Nonstop

languaging is the process of tracing my thoughts through language, simultaneously in spoken and written form, in real time, in front of a spectator. In other words, I write and talk continuously and at the same time, articulate my thoughts as they appear. In nonstop languaging, the gaps between words do not exceed the milliseconds a breath lasts or the millimeters of blank space between digitally written letters. The practice aims at synchronizing the action of thinking with the articulation of thinking, and it is always battling with the question: *How can I witness my thoughts at the same time as another person?*

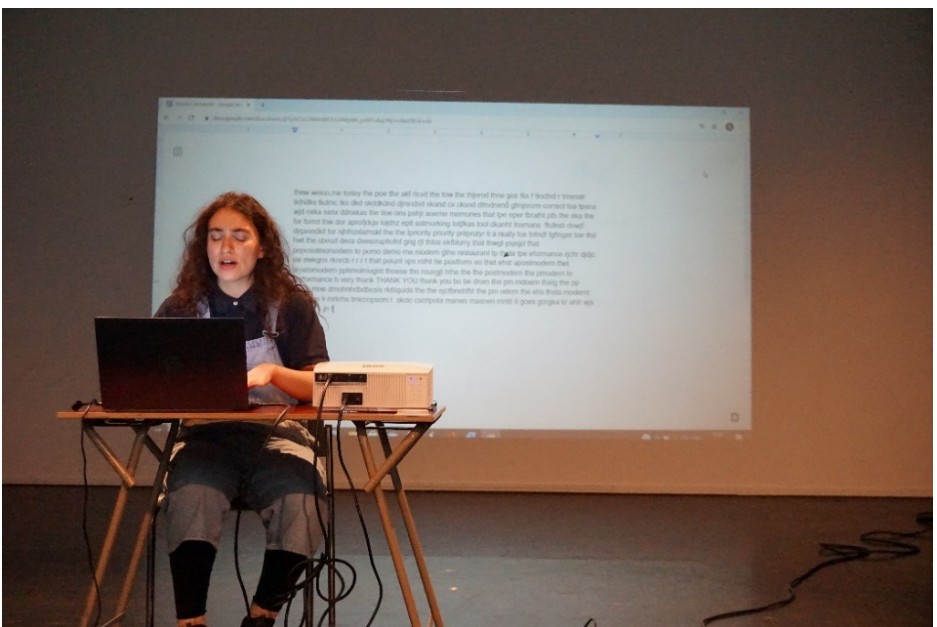

**Figure 1.** One of my first performances of nonstop languaging at HOME OF PERFORMANCE PRACTICES, in early 2021. In this experiment I simultaneously talked and wrote nonstop, projecting the live writing behind me. I invited the audience to witness the performance for 20′, followed by a feedback session. Photo: Ella Tighe, ArtEZ University of the Arts, Arnhem, 2021.

The aim of my research is to contribute to a critical mass that foregrounds feminine writing (from Cixous' *écriture féminine*) and renegotiates the confinements of knowledge production within artistic academic discourse. Luce Irigaray, in *This Sex Which Is Not One*, draws a parallel between fluids and the voices of women. In the chapter "The Mechanics of Fluids," she notices how fluids are misinterpreted within the logic of science, and how the inability of the scientific language to explain their physical qualities leads to those qualities being disregarded as non-existent (Irigaray 1977). Building upon the example of fluids, she explains how the male-constructed language system silences the feminine voice; how that language identifies as inappropriate and insignificant everything it fails to comprehend, with ears "clogged with meaning" (Irigaray 1977, p. 113), unable to engage with the continuity and the linearity of the fluid feminine voice. Furthermore, Hélène Cixous, in *The Laugh of the Medusa*, encourages women to write, to invent a language of their own, and to "draw their story into history" (Cixous 1976, p. 881). In my performance practice, I aim to speak about the academic language as a confined space through a language confined to my own constraints, an autotheoretical language that searches for her[1] own meaning-making tools, ways of producing and processing knowledge, and ways of carrying and articulating the self, all situated in the ephemeral encounter between performer and spectator.

In this research, the performance practice and the practice of theorizing are not separated. I perform autotheory by merging methods of articulating autobiography with methods of forming and digesting theory. Therefore, the main text of this work was created in moments of performance. Specifically, it was created through a series of performances of nonstop languaging during my biweekly radio show, *a continuous present you know it*

*no, is a continuous present*[2] (2021–now), at radio WORM, NL. The process started with a pre-research phase, in which I collected references, notes, and citations in a document, and then reworked these through nonstop talking and writing over that document during my radio performances. After I transcribed the recorded audio into written text through nonstop writing, I selected fragments of the different performances to compose the main text of this article. Finally, I revisited the text that you will read here, in order to cite the references that were still present and to write the introduction.

My interest in streams of consciousness derived from the urge to always allow space for encountering the unknown within my performances. Consequently, I developed nonstop languaging as what Hejinian calls a process of acknowledging: a process that primarily entails a form of not knowing (Hejinian 2000, p. 2). Acknowledging is a mode of thinking that does not claim "to know what things are" but "to know that" they are. In knowing that without knowing what, Hejinian finds the possibility for an inquiry through language and of language itself; an inquiry that is open towards the other (Hejinian 2000, p. 2). Thinking on the notion of the other, Hannah Arendt finds that "nobody exists in this world whose very being does not presuppose a spectator" and that in life "being and appearing coincide" (Arendt 1978, p. 19). Furthermore, feminist thinker Adriana Cavarero coins the notion of *the necessary other*, whose presence, she argues, is what constitutes the narratability of one's self (Cavarero 2000). Building upon these theories, I set the presence of a necessary other (mainly referring to another human) as an indispensable parameter for my autotheoretical practice of nonstop languaging to take place.

When researching autotheory, the question of what constitutes an autobiographical process becomes unavoidable. Proposing an alternative to the retrospective character of autobiography as defined by Philippe Lejeune in *The Autobiographical Pact* (Lejeune 1989, p. 4), in nonstop languaging, the present is experienced as continuous. This notion is conceived by Gertrude Stein, who writes that "a continuous present is a continuous present" (Stein 1926, p. 220). By performing nonstop languaging, I engage with autobiography as a process of acknowledging and articulating the continuous present rather than retrospectively documenting the past.

This text, although intending to challenge the academic format, has complied with the given referencing system in order to amplify the voices that formed it. Building upon Woolf's note on the "accumulation of unrecorded life" (Woolf 1929, p. 75), nonstop languaging aims to disturb the accumulated silencing of specific (feminine, subjective, unedited) modes of working with language within artistic academic discourse, both literally (by nonstopping) and through the repeated citation of works that represent such voices. Citation, according to Fournier, is a way to define one's writing in relation to a self-determined community and history (Fournier 2021). Since I started this research, I have mostly studied the works of women. I believe that through citation, autotheoretical works can make space for a wider range of voices/modes/formats to produce and disseminate academic knowledge. Citations are bricks, which build books, which are houses; they build feminist shelters (Ahmed 2015). The shelters are built not only for the bricks themselves to be protected, to be supported, or to belong, but for anything that needs placing inside a house, which is made of walls, which are made of bricks, which are citations. Citations are bricks which build houses (Ahmed 2015). As discussed further elsewhere in this Special Issue, numerous ways of juxtaposing the personal and the theoretical have been explored both in the past and the current context. This introduction is meant to function as a tool to help the reader enter into, and contextualize, my research. It is also intended to aid in the translation of my performance practice into the format of an academic paper. Nevertheless, within the context of autotheory, I encourage you to avoid reading the main text as the auto- and the introduction as the -theory. This work intends to contribute to a critical mass of autotheoretical works which, by thinking "in and of language," search for their own methods of forming artistic academic discourse (Hejinian 2000, p. 2). Given the opportunity to exist in an academic journal, my artistic research would sabotage her[3] own intentions if the performance text were to appear as anything less than central.

## 2. A Continuous Present, You Know It No, Is a Continuous Present

There is immense political power here (Zwartjes 2019) hear eclectic congratulations, make sure to forget to look always has been a perfect elite thing (Zwartjes 2019) and it is going and going like going nonstopping citation as practice, un advertising them, particulary you the necessary other (Cavarero 2000), bus tour performance not forced entertainment rotterdam not didn't go many times, big suprise why are they running the forgotten the ear (Irigaray 1977, p. 113) fluid un clogging to you the necessary other exclusive theorizing decisions disengaged error 404 place not found ears dirty for a while you can't listen through (Irigaray 1977, p. 113) like erection viagra blue pills you can do this kind of work social media's perfect curation (Zwartjes 2019) surf and turf the chef comes from a butcher family, i don't have that, filling the voidvoice being back happy life push further behind mixing it releasing it bold letter bold assertions not necessary to arrive to filling the silence on the other side are you there>? are you there?

today we ask to be met half way (Anzaldúa 1987, p. ii) today wasn't today you get it, situated in the ephemeral encounter between spectator and performer we are not [woman has always functioned (Cixous 1976, p. 887)] insignificant (Anderson 2011, p. 86) as crucial strength order order lorde (Lorde 1984, p. 120) protected from herself (Anzaldúa 1987, p. 17) it's a banner on the street protect me from what i want protected prosecuted it arbitrary the lenses put them out otherwise you will lose you right to wear lenses if i am losing my eye at least i will have you necessary other (Cavarero 2000), tell me soemthin i can hold on too, deeper maybe that makes more sense shows significant different experience soundless the sound is picking i want it to go further you are getting a cloudy situation what is her name heidi then it was not cloudy she wears a dress and she says something about happiness i thought fear of going home (Anzaldúa 1987, p. 20) insignificant (Anderson 2011, p. 86) the autobiography the students said, i thought, pages and pages, homophpbia meant fear of going home (Anzaldúa 1987, p. 20) not really tracing the thoughts through language she sings something about happiness because the master's tools (Lorde 1984, p. 120) they mean nothing simulation of a reality going on poetics concrete poetry from right to left and i do keep that saying bracketing (Husserl 1931) the phenomenological saying so much it arrives to nothing many times she wished to speak (Anzaldúa 1987, p. 23) the odds where oddily oedipous una cultura things keep falling i would have to arrive to the conclusion the recent findings of this research suddenly we look up and it is full of mold, a home full of mold there is nothing to be done about it i can't quite remember i have always wanted

calling you back to confirm your reservation really weird experience mixing up unedited work or maybe working maybe the academic he was like why do you what is your fascination with academia the battlefield everybody hurts sometimes waking up 7am missing the pass bus the zachte-lenzen am i even writing you have lecturing as a way of presenting already acquired knowledge there is no such thing as acquired already knowledge acknowledging acknowledging acknowledging (Hejinian 2000, p. 2) he washed his face and it was clean, gonna leave the door open irigaray like the voice the water bodies of water (Neimanis 2016) confusion naming not as new thing going on fermentation (Fournier 2020) IN CASE OF EMERGENCY just be speak greek quick because i don't want to add a comment the original thing the contribution to knowledge very often however picking what's important fermentation (Fournier 2020) as a way of creating autotheory my scoby before graduation, don't, protesting other situations of everydayness chillies chillies he has a name they are playing gravel he won it why would you wear a white suit i was on fire hey mom the dress the metro 20 euros for both i can't do this and he, 1 minute radio kiki could always do that but i can't zwartjes cause there's nothing else to do every me and you projects teachers going back and fourth perspective autotheory conceptual project they often tell you themselves how to read them (Zwartjes 2019) read me like this nonstopping talking and writing at the same time she says this project teach us i like the word tutor more than teacher i like the word autotheory more than i like the longer version more than love and anarchy[4] tracing thoughts through language to write to write to write what they feel must be told (Soriano 2018) skills kills feminist killjoys (Ahmed 2015) i like the blog

but i also like the book haven't ready it bell hooks asking everytime are you ready? never gonna be ready if you ask me partner never gonna be ready, negotiating the boundaries[5] (Cocker 2016) critical mass REWORKING REWORKING REWORKING it's not a floor it's a mouth we have this deal now i am going to vacuum

there is this new song by anderson and mars also the stock went down on today's market news totally forgot about this not alone he said marriage are you ever alone and lonely totally two different things in a matter of thoughts works of autotheory are autobiographical texts (Zwartjes 2019) symbiotic culture she said happy mother's day to your scoby foregrounding the accent the cimposoion write composition and you will be tax free prime time 9 oçlock the ear (Irigaray 1977) finally found this trick to do with the letter the guy with the cardboard stay with me t.s. who is that who walks ont he other side of you (Eliot 1922) just go for it the theoretical issues by nonfiction authors kazim ali (Soriano 2018) multipilicty of voices in a way of fermentation orange wine you ferment the skin and grapes and the juice results that are always gonna be the same some sounds are always coming out of your mouth in a certain way challenge that do not fall i knew it was vulglar he killed his dog but actually he killed his dog and his wife and he is in prison now literary breath your shoulders it's not gonna be nice i want to crack it open like a bag full of natural wine butterflies i think they live only one day shaking the apple tree (Ramshaw in da Silva 2015, p. 2) let me do what they told you according to irigaray, books continue each other (Woolf 1929, p. 67) now here no-where books continue each other (Woolf 1929, p. 67) arrive at the borderland (Anzaldúa 1987) re-cap re-operation re-ellaboration characteristics of fluids (Irigaray 1977) to interpret them according to irigaray the anonymity not breathing nonstop talking laughing coughing bad poetry visual representative performances have al-ways, you understand

maybe i just like everything fournier does what do you have 4 okay 6 okay i won not really question the difference between autofiction and autotheory situated in the ephemeral encounter it carries (Cixous 1976, p. 889) the present which is continuous 22 of may the final of the final 4 everyone belgrade but me here maggie nelson she is behaving as if you know until you know (Zwartjes 2019) the ears (Irigaray 1977, p. 113) are locked one questions about autotheory: what is it to write through the body[6] (Zwartjes 2019) citing citicing how to do it ex-citing how do do it to do it losing the ear (Irigaray 1977, p. 113) the cookie crumbles in your arms why no one plays the smiths in manchester have you made a reservation we had other obligations box box box max and something about the circle the fluidity listening to my ear if i lose my ear technical issues not so much about losing the ear (Irigaray 1977, p. 113) it is about unclogging it[7] there is a reason i didn't do it for a long time, you and me the necessary other you don't need to change the text you need more engaged readers (Zwartjes 2019) will you become more engaged?

since when are you so interested in sports who are you not the same thing terrible, terrible things things happening to her body (Boyer 2019b), anti nausea meds and anciety acknowledging (Hejinian 2000, p. 2) as a mode of thinking back them if it was something else to recognize difference as a crucial strenght (Lorde 1984, p. 120) to know to not know shaking the apple tress (Ramshaw in da Silva 2015, p. 2) i though same page how apt, fear of going home (Anzaldúa 1987, p. 20), it is the first time i actually payed attention to the lyric they fall in love and end of story happily ever after never never never according to the hierarchical value of the elements the feminine i would like it to be one big thing misunderstanding reflects my language in the borders infant language (Anzaldúa 1987, p. ii) what is it called when you think you are more people than one, one body, i hate doodles, hate is a big word i dishlike them very much, all your availability how could you ever books continue each other (Woolf 1929, p. 67) coughing and burping archaeologies of the now is contem[porary with us (Lucas 2013) my ear is back the whole delay was for a good reason will never dismantle (Lorde 1984, p. 120) going to behave like you know what i mean until you know what i mean (Zwartjes 2019), engaged readers (Zwartjes 2019), the prohibited and the aproppriated

i don't know this song i would sing it for you if i knew it the body is crushing if the text is crushing because the text through the body (Zwartjes 2019) in chapter two mechanics of fluids she says about the ear[8] (Irigaray 1977, p. 113) you frame it in order to make it possible for someone else (Zwartjes 2019) like repetition in terms of refuge shelfter last shelter jeanine berlin how hau you just go for it nonstop as a the creation of a community (Fournier 2021) critical mass it's a mash she said the show must go on we had a problem i cannot pronounce it so are we going to do this namedropping: ahmed, a continuous present is a continuous present stein (Stein 1926) mother scratching through the body when you go through it come on just don't give up it's a motivation i said and hearing these words from my mouth was like theorizing 6 streams of consciousness and they make a play waves woolf (Woolf 1931) accumulated unrecorded life (Woolf 1929, p. 75) essence lack of woolf you know it she said do you ahve anymore up do date literature lucy ellman ducks and breworth (Ellmann 2019) forgot it one sentence or 8 sentences (Preston 2019) a,re you lonely theorizing is a different thing reworking renegotiating once you the boundaries (Cocker 2016) fermented you can't go up faking it until you know it do not do that replacing ephemerally fermenting (Fournier 2020) sounds a little bit different today hotel i love google your mind the foreground is the longest word she says h but i say ha reworking the boundaries (Cocker 2016, p. 14) avoiding the boundaries between the creative and the academic creative (Margeaux and Sayers 2017) no you can't listen to you borders are set to define places (Anzaldúa 1987, p. 3) so overwhelming it is almost unpleasant making sure to finish the sentce what is a difference between a sentence and a phrase η τελεία και η παύλα είναι δύο πολύ διαφορετικά πράγματα

theory presents itself as because tracing down the text it changes appropriate insignificant writing anzaldua was before borderlands (Anzaldúa 1987) borderlines border boredrom he says i'm gonna make a project about boredom you have to start from something that motivates you free form from the body that transforms ferments IT IS A METAHPHPR metaphor not stealing everything she ever wrote plagiarism the house is clean don't stress hey love how are you reciprocity what is your brother doing in southampton there is tank it has gases censors stopping for ever rublev wrote in the camera no war and i have not heard from him still making something about it because the scoby ferments transformation material if you ask me merry christmas it's okay to take a break that is finally language as a confined space it is changing and she said i get it the dyslectic text you don't have to regret it or change it for me which more than two strings to do not send to the uk a plug that is different things what do you think unamplified means this number keeps calling me it was a present for nils she said i bought a tent language as a container it's pink fairytaleý kind of way i understand it annoys you bracketing (Husserl 1931) do not give up 70's french feminism cixous irigaray there is a storm in my head and it needs to rain men hallelujah it's a process that both begings with and continually engenders (Fournier 2020) there is a storm happening in my head forgot to wear a watch (on the hand) clock (on the wall) these things would stay the same no matter what the body does

transgressive writing is not necessarily about sex (Glück 2016) pov person of interest is about articulating the present new narrayove (Glück 2016) my brother had a party my mom comes in the morning and then we find a hickey the organ the brain is an THE BRAIN IS AN ORGAN PART OF THE BODY how does she write the name bet without remembering who the person is maybe it's all a blur now i see you have been getting behind your notes the cables the airwaves the weather today would be terrible but keep hoping faking it till you making it you are so vague aerostol that you wear and you lose gravitiy continually engenders vital matter (Fournier 2020) for you information andaluzia is a place anzaldua is a writer was? fermentation is not a way of rottening but a way of changing transforming capitalist terms i beg you begging you for mercy new narrative linear non linear narrative in sync imagine if we were starting one two three i dare you start repeating after me there is a light that never goes out finding a juicer out of ceramics there is a juice you mix it with sugar naturally tracing thoughts through language felt empty for a second what do you say then when it taste smells feels nonsense you can use it for vinegar i am ignoring it because

when i take it out i see realize it is dead stainless steal i am stainless and my certain is still for a rapper www.com that wou;d be a nice website

the incorrectness aims to unclog you your eye your ey ear (Irigaray 1977, p. 113) if only it was possib;e to keep going forever like rottening for a is she coming the uncomposed you are so uncomposed for a dead frog in the bottom of a lake dishonesting stopping autocorrect offf maybe it should be on where else is it to go from here the acummulated silence they ask a state and then a city like when they say you travelled to marousi you had so many other beautiful words you could have chosen theorizing i say with assertiveness selfish this was from you the voice is different never caughed before through the body you know falling apart we just wanna have a glass of wine inside of you they made you a favour it is a container needing to fight through the nonsense BRIDGING THE NONSENSE WITH THE SENSE it is more the fear that you find it nonsense don't want to spill your find kill your job you have an issue it is a special issue work your internal issues

maybe you have to become a freelancer to become an artist intentionally why do you repeat if you do not want to correct if you are here thank you oh you are from greece prego this is not greek it's italian what was greek for thank you ευχαριστώ and then they say it wrong the container the confined space which cannot ever be reworked maybe you will maybe it will you as academia do not give up waking up doing this today AGIAIN life long commitment if i was to set a theme it would be commitment every drop that falls from the dish she killed her kiljoy there is a hair inside my food and it's not an animal it's a person repeat after me IT IS A METAPHPOR write it do not take it literaly unliterary articulating the present but there was no language so we made one (Glück 2016) jeanine was saying the moment you articulate it it goes away i disagree do you want a fluitje or a biertje keep getting scared because if it was fleshy as word the necessary other (Cavarero 2000) i would mind because witnessing (Arendt 1978) because there is no such thing as not witnessing there is no such thing as an arc of this linearity renaming titles he made many tiles it was messy the unconditional ;ove the tongue keep your back straight because it's bad because the grandfather my grandfather is dead do you want to be buried or burned dark matter speak bitterness (Forced Entertainment 1994) really long table 6 h 6 streams of consciousness the waves (Woolf 1931) i would like to read it a continuous present (Stein 1926) not one of them stopped living ever multivocal perspectives history that is not participatory i didn't get it then and i did not get it now wish sometimes to arrive to ideas you multivocal bubble who is that on the other side of you (Eliot 1922) but who is that the wasteland april sis is the cruelest month (Eliot 1922) if you ask me there are other things that are cruel morning sickness when you are pregnant i don't know what the baby has to do not mine but just saying might be a little bit confusing fore some fore grounding number 4

bracketing (Husserl 1931) now, thinking about reality, she says, reality is all there is she says (Hejinian 2000, p. 8) enormous i think huge is a better word i'm just a girl in front waiting to be loved continues on the street blue shirt they go he has to hide the house the best condition of the house ecompassing why this word came i was speaking french without knowing it there were those who hold scores penalize miscarriage fucking scary that enormous machine when i say page and when i say chapter i feel like a chapter, nonsense why don't ypu stay with me vote for your favourite player vote for your favorite cruelest month, insignifant (Anderson 2011, p. 86), and you i can't quite remember what the markers are that mark as (Boyer 2019b) and who we are when we are not language itself

hey, breaths there is immense political power here (Zwartjes 2019) in electing fosse was born and now she exists in the world not born but conceived arriving to situations theory has been moving the head the perfect curration (Zwartjes 2019) of the perfect moments red book eirini kartsaki book (Kartsaki 2017) the jukebox when in new york song and wait for ages joy division it is not this is not a love song this is a heavy metal song vs academic writing vs creative writing (Margeaux and Sayers 2017) responsiveness in the situation middle aged people you are my dad he is still conjuction cutting the thing not liking the sound in my ears the challenges of writing and publishing (Margeaux and Sayers 2017)

wanted to give him a chance to answer forgetfull forgiveness forgetful been kicking have been forgetting to text the mouth has to warm up curation like a challenge these projects move this microphone on the wrong position not on the wrong side an outdated division (Margeaux and Sayers 2017)

however these projects when i am reading sharpe preciado testo junkie nelson says something about assertions clearing out all uncertainty (Nelson 2015) something is pulling my hair down sometimes sexy the field of autotheory multiplicity of form (Soriano 2018) for something to be multivocal without being participatory unclogging the ears (Irigaray 1977, p. 113) like perioxide you have to understand it exists there the phallus i graduated by the way me me me jeanine goes to the window (Durning 2010) the best performance i've ever seen online and in person speak bitterness forced and entertainment in hhhhau (Forced Entertainment 1994) the memory is still here short term and long term two different things i know we keep saying this we we two different streams imagine six if you are listening necessary other sara ahmed and hannah arendt confusing them one said citation as a shelter (Ahmed 2015) of community (Fournier 2021) reworking the boundaries (Cocker 2016) reworking stimulating out of the red and not out of the blue double backhand shorted 3 o clock, and the other one about witnessing and living (Arendt 1978)

feminist architwcture builds are houses bills are houses belastigndiesnt envelopes citations build house, are bricks, citations are bricks which build houses (Ahmed 2015), this is a payment reference, on purpose the mistake you want to know my bsn number? to go back to some contract the tiny green shoots (Anzaldúa 1987, p. 91) children from 7 to 11 continues present continuing continues the presence continuing presences (Woolf 1929, p. 95) the same way that she says all kind of things like this, terrible things are happening to her body (Boyer 2019b), or never i am kind of lucky in that term came back from 2 sets down fought back single handed backhand the new generation the future of italian tennis, acknowledging (Hejinian 2000, p. 2) is a mode of thinking, same like tracing shaking the apple tree (Ramshaw in da Silva 2015, p. 2) it does not claim to know (Hejinian 2000, p. 2), knowing and not knowing not as the opposite (da Silva 2015, p. 7) i don't know why i don't know why one second and be right back should make an intro and i dare you i dear you how do you googling how to network find the red thread in your work to sneeze is a different thing than to cough just so you know what to underline books continue each (Woolf 1929, p. 67) persist insist books continue each other (Woolf 1929, p. 67) i am typing acknowledging (Hejinian 2000, p. 2) it's a mode of thinking that does not claim in knowing what without knowing that (Hejinian 2000, p. 2) is a mode of thinking to know that things are (Hejinian 2000, p. 2) you are speaking so fast in my head it's softer hejinian she betrays him and says it's kismet language itself hate it when i say and then, it keeps coming back the switching of codes in this book (Anzaldúa 1987, p. ii) going to say then 1000 times until i never say it again searching for we inside the text why this felt releveant she means a totallly different thing let the sunshine in miss you very much how i wish from that album how i wish how i wishing don't make it cause it might not come out allowing texts autobiographical interogating the situations the repetition your voice the base 5 minutes and then it snaps i hope my tubes fallopian are not dry wikipedia the human body read me read me I TEACH YOU HOW TO READ ME READ ME (Zwartjes 2019) ready or not there is not really before because there is the not-yet and the no-more in between the present (Boyer 2019a) fos means light in greek i think she knows

this soong the one you already know maggie nelson talks to you like you know until you know (Zwartjes 2019) make a sentence with the word understatement text us your sentences having the clocks dancing when he is playing 5 times in the gym a pen inside my head scratching potentialy one side and the other i am going to talk like this until you understand me scoby mothers on mother days what if my scobies are moldy failure emtpty from the inside someone broke a glass over my bed everyone comes up with all different things pushing the boundaries i never meant to hurt you cAUSE AIN'T NO SUCH THING AS HALFWAY TEXTS the mother being the scoby you put something and you don't do something a yeast a symbiotic a cite citation as a shelter creation (Ahmed 2015) you is it we

or just me and you again also good no thing not everything inside of you is yours (Boyer 2019a) because nothing the person is shaped augustinian recollection the no more and the not yet (Boyer 2019a) the continuous present is a continuous present (Stein 1926) a rose is a rose (Stein 1922) interiority the difference between the auto of auto-pilot and auto- of autotheory (Boyer 2019a) the autobiography of a;lice (Stein 1933) haven't seen you in years hope this email finds you well bracketing (Husserl 1931) vulgar you have an american passport introspection retrospective autobiography should not take place in intro-retrospect this wooden stick about your ear do not get it repeat after me not everything inside of you is yours (Boyer 2019a) i can see you she said you are fired today girl wind from the side ai wei wei was he really in exile installator i'm installing sculputeres do not google it you are going to use the passport everyone can watch the match be there for his birthday why are you going away from greece special moment for you now too the best best best ever how little did i know then inappropriate the anonymity of the radio sometimes a scary thing it was you remember? when i said soemthing in the train you were calling me songs on the phone i had put this fairytale gone bad from 2000 crying from the possibility of losing those girls from your life foregrounding sunday sunglasses best performance i have ever seen something about ex boyfriends as if they are many men don't look to the sky no more lyrics loved the halftime show 5 h from here more and more expensive texts tests in the magazine a horse a dog a duck horsed entertainment keep conusing them no matter how no matter what not everything inside of you is worse

making the space available because it makes it possible for someone else to do it, naming it (Zwartjes 2019), it is alright to write autotheory because fournier 2021 book mit press (Fournier 2021) exhibition around the neck what present do you get to a baby sometimes you want to call your mom and say what do i do no way how little irrelevant in new work going away never coming back 10 years almost i said no way amsterdam sign hate they should turn it in two sides one i love one i hate there are two sides in every lie try it out and see what happens what if she wants to be an artist and it's all on me bracketing (Husserl 1931), this method entering phenomenological situation bracketing (Husserl 1931), encounter because same space at the same time keep hearing this sound of a paper irresponsible if you ask me which you don't have you considered silence getting at the top of my voice foregrounding in augustinian recollection feminine voice to do the audition i don't like to talk about the past because

institution (Ahmed 2014) typically refers to not a person but the organism when i am saying the white men is an institution (Ahmed 2014) the persistence mechanism institution in you, wash some dishes i do not have a dishwasher by hands anymore you lose the ability to lose wash things by hand, tracing thoughts through language tell me something i can hold on to, boy you re something else, went into the sea praise of risk (Dufourmantelle 2019), you have thrtee minutes in praise of risk (Dufourmantelle 2019) is this an expression? technically going to get dementia already knocking on wouud preoccupied and wing wing it's always a bit confusing what do you mean with that wing how many hours do butterflies live horrible advertisment of a whole country on tv here you buy it greek easter can we please write a review he never says please jen next gen confusing opelka with alcaraz unecessary life decisions he says if you are not sure about your life's decisions do not put it on her, he generalised and what did we say about generalizing?

pointless to correct, how does a reader become an engaged reader (Zwartjes 2019) by being a necessary other (Cavarero 2000) in an ephemeral performance of nonstop languaging like inging (Durning 2010) the end the pill the spiral the contraception 5 years and then mwah mwah it could have been other things but it's over now moving contact improvisation retrospective act of forgiv eness in the past 15 minutes are impatient push a little further rework the boundaries (Cocker 2016) unexpectedly why are you not proffesional you were just born if you have a problem with this do not put it on other people are you following he says why do you speak so fast hey guys is this a dictatorship or a directorship 1994 and 1992 copied his email pattern the sunday gospel hitting the legs the hands the toe the nose the automatic font of email is calibri would prefer he;vetica arial geneva why

do they call them after cities more like funny switching citiers the watch on the hand retrospective agua viva (Lispector 1973) carries (Cixous 1976) the present claricelispector smoking a cigaret on the video there is two sides in every coin actually there are three there is the one where it is rolling nothing better than being driven by friends in the airport and listening rolling rolling rolling on the river going 3 h in advance thinking it is necessary contributing created the brain's patterns highlighting them nonsense girl keep running 100 m what is your score 3 2 1 get in a line everyone ready 3 2 1 and we are racing in the netherlands are you there yet?

an inquiry in and of language (Hejinian 2000, p. 2) the fire the structiure of language the isntitution of white men (Ahmed 2014) not the person, the language of the borders (Anzaldúa 1987) the planet women stupid book the appropriate who made this kind of work it would be so much easier knowledge is fine i like acknowledging (Hejinian 2000, p. 2) more than knowledge build around the masculine (Cixous 1976) the history the mars the women the history the unrecorded life (Woolf 1929) the critical mass the tell the bell the books the instituttion the anti nausea meds (Boyer 2019b) forgot how to say it in english that is why i say betterschap there is this button, who would be crazy enough to write the whole world in a button, the thing to know the not knowing (da Silva 2015) the if you write the poetry foundation the boyer kept thinking it was broyer the failing not distracted our time flew by nobody exists in this world whose being does not presuppose a spectator (Arendt 1978, p. 19) the betalings number one stops earlier actually what is inside the belly he explained the whole process they take the liquid of the eye the same with abortion i guess vacuum cleaner is a thing they use a lot in surgeries it falls back drops back into position you have to stay for a month hope i pressed the button the theorizing the acknowledging (Hejinian 2000, p. 2) i thought how apt noting as a way of fermenting (Fournier 2020) the underlining, the cruelty of the month the falling back into the waiting for the end search, the pressing the letters really hard.

**Funding:** This research received no external funding.

**Conflicts of Interest:** The author declares no conflict of interest.

## Notes

1　Language in Greek is a woman.
2　Please visit the link with excerpts from my bi-weekly radio performance at WORM: www.antriannamoutoula.com/radio accessed on 20 May 2022.
3　Research in Greek is a woman.
4　For further inquiry and exploration into the practice of nonstop languaging, please visit the online archive of encounters with necessary others in the weeks before my graduation performance at HOME OF PERFORMANCE PRACTICES, ArtEZ University of the Arts, NL. Accessible at www.antriannamoutoula.com/ilikethelongerversion, accessed on 28 October 2021.
5　As writer and artist Emma Cocker says, "to know your limits does not mean to dutifully remain within their bounds but rather… to develop the border knowledge that will allow the limit to be negotiated differently or rendered porous, to learn where the boundaries are and be mindful of how to facilitate their crossing" (Cocker 2016, p. 14).
6　Forming a space for the stream of thought to appear through its own confinements (grammatical, syntactic, citational, etc.) within academic discourse, is the way of "shaking the apple trees" (Ramshaw in da Silva 2015, p. 2) that I insist on. Nevertheless, for the apples to fall, there needs to be more than a single act of shaking. It requires "continuing presences" (Woolf 1929, p. 95), continuously shaking the apple tree with different shaking methods, different rhythms, different intensities, different understandings of what shaking means.
7　The autotheoretical knowledge that nonstop languaging and other similar practices work towards requires engaged readers (Zwartjes 2019) with ears able to listen, allowing space for a variety of forms, structures, and linearities. In order for those to exist, autotheoretical practices have to unclog the ears that are "clogged with meaning" (Irigaray 1977, p. 113), and challenge the "deaf male ear, which hears in language only that which speaks in the masculine" (Cixous 1976, p. 881).
8　When citing "the ear," as Irigaray describes it, I am referring to the ears clogged with meaning and "closed to what does not in some way echo the already heard" (Irigaray 1977, p. 113). On the other hand, "my ear" refers to my physical ear and its different states during performances of nonstop languaging.

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
