# Peer review of "Unclogging the Ears: Nonstop Languaging as Autotheory in Art and Academia"

_arts, 2022_

Round 1

Reviewer 1 Report

Dear author,

thanks so much for your text. When I saw it brought autotheory together with streams of consciousness and nonstop languaging, I was eager to get right into it. Before I crack on, I wanted to make a note that I found it difficult to evaluate such alternative work with the quantifying tools available for traditional peer-review. I am glad I have this space to nuance those.

As I was just saying, I was so genuinely glad to see a proposal that wanted to explore stream of consciousness. And your references were spot on. The idea to write in a continuous present is fruitful and exciting.

Your proposed text raised so many questions for me. On the one hand, I appreciate your practice and the fact that it is a well-informed practice, too. On the other hand, I find it hard to translate into a paper. This is what I will try to elaborate on.

The questions that frame my review go from the unavoidable ‘what is autotheory’, and ‘why autotheory’; to ‘what gets to be autotheory’ and ‘why do we write (and engage with) peer-reviewed papers?’

Let’s take autotheory to be, at its core, “the integration of theory and philosophy with autobiography, the body, and other so-called personal and explicitly subjective modes” (this is from Fournier’s book). I think your proposal does this in an original way: there is an explicitly subjective voice that is quite literally trying to digest theory.

As Fournier herself demonstrates, autotheory can be transmedial and multidisciplinary. But (my point, not Fournier’s) - different mediums have different strengths, aims, and contexts. A performance can be autotheory. A paper can be autotheory. A paper can be performative. But can a performance be (or rather, become) a paper? How to translate a proposal of this type, into what a paper seeks to do?

A paper is a tool for sharing knowledge. I am all for pushing the boundaries for what that can look like, and what is legit to include in one. But I am afraid I read your proposal as an annotated performance. That is a format that can be pushed further, but as it is now, to me, the proportion between reflection, or theorising, and experimentation is not quite balanced in a way a paper needs to be.

I have the impression that for you it was important to not ‘break’ the languaging into chunks that would be analysed in a paper, or to use sections of it to illustrate a point. Hence, the whole stream, plus footnotes. I get it, I get it is intentional. But for a reader, at least in my experience, it still reads as such: the footnotes are the paper. And they are interesting! They hint at having plenty to say. But they barely have any space to do that. I really wish I could read more of what was beginning to be hinted at in them.

In other words: precisely because of the processual/circumstancial particularities of how the languaging practice is presented (and present-centric), the theorising happens sparingly, off-site, in the footnotes (retrospectively, too). Yes, there are plenty of theoretical references in the ever-moving body of the text, but the digestive gurgling makes them more like preformative devices than theoretical ones (a practical, methodological question I have here is how, if the text is all unedited, the high-speed languaging allows for all of these references - are these off the top of your head? Do you write and speak and consult books? Do you have them ready to copy-paste? I am genuinely curious). 

The other conflict I sense in the paper is that the sparse theorising in the footnotes can read as defensive, as if they are designed foreseeing criticism such as the above. They shield the proposal, reflecting any doubt on whether the paper works as an academic paper by suggesting that those doubts come from a disengaged reader (footnote 8), a reader that only reads ‘masculine’ (footnote 8), a reader that does not care to challenge the silencing of some voices within academia (footnote 10), or one that is happily settled within the confinements of discourse (footnote 11). And yet - I am not sure the mini discourse being formed in the footnotes escapes those ‘traps’ itself: at the end of the day, the way it sounds, the way it uses references, the way it relates to the point it is trying to make… is typically academic. Just heavily minimised.

So, what would I suggest from here? Again, I want to be clear that I enjoy the seed of what you propose, but, for me, there needs to be a step that re-balances the proportion of its different parts, or of its different tenses, if you prefer. If the following questions help in finding what that looks like, please take them: What is your point? What do you want your reader to learn? Where does your reader learn that? In what tense is that point being made? What is the difference between your performative practice and your academic writing? How does framing your performance in the context of a peer-reviewed journal challenge your practice and the way you communicate?

I hope some of these can be generative. I am very much looking forward to reading whatever final shape you decide to give this piece. Thanks again for the challenge and letting me think alongside your work.

Author Response

Dear reviewer,

Thank you very much for your thoughtful feedback, the time and attention that you dedicated to my work. While reading you comments I felt that you have taken into serious consideration the intentions of this proposal, and that you allowed the space to truly think alongside my practice. I very much appreciate that.

Please find attached a summary of the revisions I have made to the text, as well as a list of responses to specific aspects of your review.

Reviewer 2 Report

This is an interesting piece which has much potential, especially as a critique of 'traditional' academic writing. However, several of aspects of it gave me pause.  I have divided my comments between ones deemed 'important' to address before this can be published, and 'suggestions/further questions':

Important:

a. It sounds as if this writing is meant to be performed, and/or has been written during a performance: where and when did this performance take place? Further context is needed and the dimension of this work as a performance needs to be padded. 

b. The notion of 'languaging' needs further explanation. There is much potential to develop this concept further in relation to autotheory. Is languaging a form of speech performance and if so, what is its relationship to writing? What is languaging's relationship to stream of consciousness (I am thinking of the reference to Stein in particular)? 

c. I wasn't entirely sure the piece did justice to the works and passages referred to. Is the point to juxtapose these works? As it stands, these works do not talk to each other but seem to merge or be absorbed within the languaging process. This doesn't always feel right given that many of them are from marginalised writers who use creative-critical forms of writing to make their political voices heard.

Suggestions/further questions:

d. If this performance was recorded, could the recording be published together with the writing?

e. What is the relationship between the different language(s) spoken in the piece? What is the author's relationship to English, and how does it matter in the context of languaging? 

f. Theories of "translanguaging" might be of interest in the context of this work (or moving on from it). I also recommend the book Thinking Through Relation: Encounters in Creative Critical Writings (Peter Lang, 2021) which contains several creative-critical essays on or about languages and borderlands.

Author Response

Dear reviewer,

Thank you very much for your feedback, the time and attention that you dedicated to my work. Your comments have supported the translation of my practice into the format of an academic paper and challenged me to further develop and contexualize my research.  

Please find attached a summary of the revisions I have made to the text, as well as a list of responses to specific points of your review.

Round 2

Reviewer 1 Report

Dear author,

thanks for this new version. I appreciate how receptive you where to my original concerns. I am now even more intrigued by your practice than I was at first!

While I remain slightly unconvinced that the direct translation performance-to-paper is the best way to go about "infiltrating" academia with other possible languages (a cause I am genuinely supportive of!); I acknowledge that this is a personal take, and that other autotheory scholars may be more adventurous than I am.

So, thanks again for the more thorough contextualising and clarification. I am happy to recommend publication of the piece as-is.

Take much care and thanks for the lively exchange!